# The Membrane Proteome of Spores and Vegetative Cells of the Food-Borne Pathogen *Bacillus cereus*

**DOI:** 10.3390/ijms222212475

**Published:** 2021-11-19

**Authors:** Xiaowei Gao, Bhagyashree N. Swarge, Henk L. Dekker, Winfried Roseboom, Stanley Brul, Gertjan Kramer

**Affiliations:** 1Molecular Biology & Microbial Food Safety, Swammerdam Institute for Life Sciences, University of Amsterdam, Science Park 904, 1098 XH Amsterdam, The Netherlands; x.gao@uva.nl (X.G.); b.n.swarge@uva.nl (B.N.S.); 2Mass Spectrometry of Biomolecules, Swammerdam Institute for Life Sciences, University of Amsterdam, Science Park 904, 1098 XH Amsterdam, The Netherlands; h.l.dekker@uva.nl (H.L.D.); w.roseboom@uva.nl (W.R.)

**Keywords:** *Bacillus cereus*, spore inner membrane, proteome

## Abstract

Membrane proteins are fascinating since they play an important role in diverse cellular functions and constitute many drug targets. Membrane proteins are challenging to analyze. The spore, the most resistant form of known life, harbors a compressed inner membrane. This membrane acts not only as a barrier for undesired molecules but also as a scaffold for proteins involved in signal transduction and the transport of metabolites during spore germination and subsequent vegetative growth. In this study, we adapted a membrane enrichment method to study the membrane proteome of spores and cells of the food-borne pathogen *Bacillus cereus* using quantitative proteomics. Using bioinformatics filtering we identify and quantify 498 vegetative cell membrane proteins and 244 spore inner membrane proteins. Comparison of vegetative and spore membrane proteins showed there were 54 spore membrane-specific and 308 cell membrane-specific proteins. Functional characterization of these proteins showed that the cell membrane proteome has a far larger number of transporters, receptors and proteins related to cell division and motility. This was also reflected in the much higher expression level of many of these proteins in the cellular membrane for those proteins that were in common with the spore inner membrane. The spore inner membrane had specific expression of several germinant receptors and spore-specific proteins, but also seemed to show a preference towards the use of simple carbohydrates like glucose and fructose owing to only expressing transporters for these. These results show the differences in membrane proteome composition and show us the specific proteins necessary in the inner membrane of a dormant spore of this toxigenic spore-forming bacterium to survive adverse conditions.

## 1. Introduction

The spore-forming organism *B. cereus* is a human pathogen and is known as a causative agent of food-borne illness. The organism synthesizes toxins which lead to diarrheal and emetic syndromes [1] once consumed. Bacterial spore formers such as *B. cereus* are a main challenge in the manufacturing of stable and safe foods. The spores *B. cereus* forms are highly resistant to heat, chemicals, UV, desiccation and radiation, making them hard to inactivate during food production. Their thermal resistance is, for a part, due to the presence of dipicolinic acid in the core, where it replaces, to a significant extent, water [2]. Partially, the high resistance is also a result of the spore’s multi-layer structure, with each layer adding to the protection of the core, where the macromolecules essential to live are localized surrounded by the inner membrane. Starting from spore core, the layers are the germ cell wall, the cortex, the outer membrane, the coat and the exosporium (Figure 1a). Upon sensing favorable environmental conditions, spores will return to the vegetative stage via a process called germination and lose their high stress-resistance properties. The signal inducing germination is sensed and transduced by germinant receptors (GRs) located in the spore’s inner membrane. The spore inner membrane is a strong permeability barrier to small hydrophilic and hydrophobic molecules [3]. The integrity of the inner membrane is required for spore stress resistance. Compressed and under low-fluidity in dormant spores, the inner membrane may expand up to two-fold in the first minute of spore germination without ATP production and without any new membrane synthesis [4,5]. The spore inner membrane also acts as a scaffold for the assembly of various types of membrane proteins. Apart from germinant receptors, other proteins involved in the germination process are also localized in this layer. These proteins are crucial for the transport of (macro-molecular) metabolites, the catalysis of reactions and the transduction of environmental cues, i.e., “information flow” [6]. Some inner membrane proteins are crucial to the life cycle of spores, especially for germination. Moreover, membrane proteins make up 60% of drug targets known to date [6]. After germination, the inner membrane from the spore develops into the vegetative cell membrane. By obtaining a deeper insight into the composition of the spore inner-membrane proteins, and how it compares to the protein composition of the vegetative cell membrane, we can gain new insights into the *B. cereus* spore and how it survives adverse conditions and regulates its eventual germination. This in turn can help us to identify targets to inhibit this process, which could be of use in the development of strategies to prevent food spoilage.

Mass spectrometry is a powerful tool for the identification and quantification of membrane protein composition [7]. However, membrane proteins are mostly hydrophobic and less abundant compared to the largely abundant spore coat and cytosolic proteins [8]. To overcome this, a membrane enrichment method (Figure 1b) is used, combined with label-free mass spectrometry to analyze the *B. cereus* spore inner membrane and vegetative cell membrane proteome in detail. Following bioinformatics predictions, the predicted membrane protein levels are compared between the spore inner membrane and vegetative cell membrane. Remarkable differences in membrane composition between these two very different life phases of *B. cereus* are found. This shows both how the spore inner membrane has communality and is unique in its make-up compared to its cellular counterpart.

## 2. Results

### 2.1. Identification, Prediction and Functional Classification of Spore Inner Membrane and Vegetative Cell Membrane Proteins

We analyzed both vegetative cell and spore lysates as well as enriched membrane preparations from spores and vegetative cells by mass spectrometry. We identified 1275 proteins in vegetative cell lysates, 1463 proteins in spore lysates, 1591 proteins in cellular and 1254 proteins in spore-membrane preparations (see Appendix A). The proteins identified were classified by three different algorithms, Locate P [9], PSORTb [10] and TMHMM [11], for possible membrane localization. Using this information, the percentage of proteins predicted to be membrane proteins is shown in Figure 1d. Clearly, an improvement of cell membrane proteins can be observed in the cell membrane preparation. On the other hand, the increase in predicted membrane proteins in the pore inner membrane sample was much smaller. However, of proteins only identified in membrane-enriched samples (Appendix A), 30% of proteins (188 total) only identified in spore inner membrane preparation and 56% of proteins (495 total) in cell membrane preparations are predicted membrane proteins. This shows the gains achieved by specifically enriching membrane fractions when studying membrane proteins.

Of the 1946 proteins identified in the vegetative cell membrane and the spore inner membrane preparations (Figure 1c), 244 (spore) and 498 (cell) proteins were predicted to be membrane-associated or integrated using the algorithms mentioned in the above. Of these 552 predicted membrane proteins, cells and spores share 190, while 54 spore and 308 vegetative cell-specific membrane proteins were found (see Appendix A). Functional classification of proteins with evidence for membrane localization from spores and vegetative cells was performed using Gene Ontology (GO) terms (Appendix A) and Kyoto Encyclopedia of Genes and Genomes (KEGG) categories (Figure 2). As shown in Figure 2, the main functions of spore inner membrane and vegetative cell membrane proteins are related to metabolism and transportation. In addition, many of the identified cell membrane proteins were involved in cell motility and signal transduction in line with previous results from *B. subtilis* [12]. Unsurprisingly, proteins involved in sporulation and germination were specific to the spore inner membrane. Studies [2,12] in spores of *B. subtili* have shown that germinant receptors are in the inner membrane and consist of three subunits (A, B and C). The enrichment method used here (Figure 1b) identified nine (three of them were identified for once) germinant receptors subunits and six (one of them was identified for once) spore DPA transportation channel proteins (Table 1). According to a genome analysis of *B. cereus*, the gene GerLA is a pseudogene and the protein GerB does not have a subunit A [13]. The channel protein SpoVA is involved in DPA transportation. SleB is an enzyme degrading cortex during germination and is anchored and inactivated by YpeB. Two cortex lytic enzymes SleB and CwlJ are present in *B. cereus* spores. CwlJ is located in the spore coat layer. SleB is present in the inner membrane layer and coat layer [2]. SleB (Secretory (released) (with CS))-YpeB (N-terminally anchored (No CS)) was identified in spore inner membrane fractions but not CwlJ, which provided evidence that SleB is located in the inner membrane layer of *B.cereus* spores. Depletion of HtrC in *B. anthracis* and YyxA in *B. subtilis* causes a reduced germination rate [14]. In *B. cereus*, we identified an orthologue (BC5458) of HtrC in the spore inner membrane fraction. Together this suggests that germination receptors and other germination associated proteins are also located in the spore inner membrane of *B. cereus* spores.

### 2.2. Protein Expression Levels of Membrane Proteins in the Spore Inner Membrane and Vegetative Cell Membrane of B. cereus

The abundances of membrane proteins in a sample can be compared by their iBAQ- values (sum of the peptide intensities protein(i)/number of observable peptides with 6–30 amino acids (i) [15]). This provides an overview of the abundance of proteins in the cell and spore membrane preparations (Appendix A). The expression level of annotated membrane proteins from the membrane preparations of spores and vegetative cells are shown in Figure 3. Many of the annotated membrane proteins were only detected in the spore membrane (54 of 244 proteins) or the cell membrane (308 of 498 proteins) sample. Of the 54 proteins in the spore inner membrane, 10 are uncharacterized, 12 are germination associated proteins (such as germinant receptors, DPA channel proteins), 15 proteins are involved in membrane transport, 10 are involved in metabolism (such as energy and carbohydrate metabolic pathways). Of the 274 cell membrane proteins, 40 are uncharacterized proteins, over a broad range of expression (Figure 3 and Figure 5). Of the remaining proteins, 153 are annotated by KEGG: 61 are transporters, 27 are involved in metabolism, 11 are part of quorum sensing and 19 are in two-component systems, five are in cell motility, such as flagellar assembly and chemotaxis.

The 10 uncharacterized spore inner membrane proteins were analyzed for homology to proteins using the Basic Local Alignment Search Tool (BLAST) to ascertain if any functions of interest could be assigned (Appendix A). Several proteins were homologous with proteins containing poorly characterized domains, among which were Q81IY1 (YusW-like protein in *Bacillus* sp. cl25), Q81FR7 (YIEGIA domain-containing protein in *Bacillus thuringiensis*), Q817B6 and Q81H33 (DUF2953 and DUF3450 domain-containing proteins from *B. cereus* and *B. thuringiensis,* respectively). While others also had homologues with enzymatic activity Q817B6 (Yvgn, *B. thuringiensis*), Q81G20 (Imidazole glycerol phosphate synthase, *B. cereus*). Five of the uncharacterized proteins had homology with lipoproteins from other organisms (Q817V9, Q816A5, Q815K1, Q81IZ3 and Q81IY8) of which Q817V9, Q816A5 and Q815K1 showed high homology with sporulation lipoprotein YhcN/YlaJ (from *Bacillus* sp. cl25 and *B. cereus*). This underscores the membrane localization of these proteins, although it does not shed light on their function due to the limited functional information available on their homologues in other organisms.

The 40 uncharacterized cell membrane proteins were also characterized by homology searching. In the cell membrane, 22 of these uncharacterized proteins showed high homology with uncharacterized proteins in other organisms or poorly characterized domains. Whereas 18 proteins also showed homology with lipid-binding or membrane proteins from other bacteria. The remaining uncharacterized proteins identified either had homology to antibiotic resistance genes in other organisms, e.g., Q81B99 (highly similar to a beta-lactamase from *B. antracis*) or to the virulence-related Tetratricopeptide (TPR) repeat protein [16] from *B. toyenensis* as was Q817L2. While Q81FN3 had homology to M4 peptidases [17], Q81FG0 and Q81AX0 had homology with proteins having enzymatic activity towards sugars (see Appendix A). As with the uncharacterized proteins of the spore inner-membrane, homology searching can reveal limited additional information of the protein function of these putative membrane proteins in *B. cereus*.

We previously explored the abundance of insoluble proteins in the spore-coat of *B. cereus* using a targeted quantification concatemer (QconCAT) approach [18], in our current study we use a label-free approach to estimate protein abundance using iBAQ values (see above). The germinant receptors are of particular interest, as these are thought to be expressed specifically in the inner membrane of spores as is the case for *B. subtilis* [19]. The seven germinant receptors of *B. cereus* are made up out of multiple subunits [13]. The germinant receptors are thought to form a large multi-protein complex called the germinosome [20]. Although, such a complex has yet to be isolated from the inner membrane of *B. cereus*. The germination-related proteins detected in the current study in the inner membrane fraction of *B. cereus* are shown in Table 1. Of 15 germination-related proteins, we could determine the expression level in the inner membrane extract (Figure 4c and Appendix A) was analogous to a prior study in *B. subtilis* [19]. Of the proteins that are identifiable as homologues in *B. subtilis*, GerD and SpoVAD (of which *B. cereus* has two) have a similar stoichiometry 1.0: 1.7–4.6 GerD: SpoVAD versus 1.0: 1.9 GerD: SpoVAD in *B. subtilis* [19]. The ratio of expression of 1.0: 2.2 SleB: YpeB in the inner membrane is not very different from the 1.0: 1.3 (SleB: YpeB) expression ratio reported for the spore coat of *B. cereus* by QconCAT targeted proteomics [18].

### 2.3. Specific Membrane Bound Proteins of Interest

Relative expression of 190 predicted membrane proteins that were quantifiable in both the spore inner membrane and vegetative cell membrane are shown in Figure 3. Of these, six proteins had a significantly elevated level in the proteome of the spore inner membrane while 63 had a remarkably elevated level in the vegetative cell membrane proteome. The six proteins that had a higher expression level in the spore inner membrane proteome were metabolic enzymes and integral membrane and adhesion proteins, while the 63 proteins with an increased expression level had a broad scope of functions ranging from transporters, cellular motility, metabolism and environmental perception (see Appendix A). Altogether these quantitative data show for the first time the communality and uniqueness of the protein’s makeup of the membranes of spores and vegetative cells of *B. cereus*. Below, several specific protein groups of interest are discussed.

#### 2.3.1. Membrane Transport

A clear membrane-bound class of proteins are transporters which bring in a variety of micro- and macro-nutrients. The phosphoenolpyruvate:sugar phosphotransferase system (PTS) is one of major carbohydrate uptake mechanisms in bacteria, that utilize phosphoenolpyruvate as a source of energy. Many types of carbon substrates are transported by the PTS system, like glucose, fructose, trehalose, maltose and so on. The transmembrane protein PtsG (BC4050) for glucose transport was common in the spore inner membrane and cell membrane. Whereas, ScrA (BC0775) mediating starch and sucrose transportation was also shared, although significantly less abundant in the spore inner membrane. In contrast, vegetative cell membranes showed a larger variety of transporters for different types of carbohydrates, like fructose and trehalose (Figure 5a).

The uptake of carbohydrates is not completely mediated by the PTS, and non-PTS sub- strates can be transported by permeases, ABC transporters or carbohydrate facilitators [21]. More transporters of non-PTS substrates were identified in cell membrane preparations (Figure 5b). In the cell membrane, ABC transporters ABCB (BC0162, BC3679, BC3678) and ABCC (BC1955) subfamilies were detected in the cell membrane only, as was one other ABC transporter BC4016. Two other vegetative cell membrane-specific transporters were BC4016, a cyclodextrin transport ATP-binding protein. Only the ABC transporter BC1927, which is active in branched-chain amino acid transport and quorum sensing was only detected in the spore inner membrane.

#### 2.3.2. Metabolism

Many of the components of cellular energy metabolism are membrane-associated or membrane-bound. The acetyl-coenzyme A synthetase (BC2489) can convert acetate into acetyl-CoA under low-acetate conditions. This enzyme is speculated to ensure enough acetyl-CoA is present for acetylation [22]. Oxidative phosphorylation is a cellular metabolic pathway that at the cell membrane of prokaryotes, using the enzymes therein and the energy released from the oxidation of various nutrients to synthesize adenosine triphosphate (ATP). During oxidative phosphorylation, electrons are transferred from electron donors to electron acceptors, in redox reactions. The energy released by redox reactions is used to synthesize ATP. In eukaryotes, these redox reactions are mediated by five protein complexes, whereas in prokaryotes, many different enzymes exist that utilize various electron donors and acceptors in the electron transport chain (or respiratory chain) [23,24]. The aerobic respiratory chain of *B. subtilis* contains Type-2 NADH dehydrogenase (NDH), succinate dehydrogenase (SDH), glycerol 3-phosphate dehydrogenase and several menaquinol oxidation complexes [25]. In this study, the NADH dehydrogenase NDH (BC4925, BC5061, BC4938, Figure 5d), succinate dehydrogenase (SdhA BC4517, SdhB BC4516, Figure 5e), cytochrome c oxidase (CoxC BC3942, CoxA BC3943, CoxB BC3944), cytochrome aa3-600 menaquinol oxidase (QoxC BC0696, QoxB BC0697, Figure 5f) and F-type ATPase were expressed both in the spore inner membrane and cell membrane fractions of *B. cereus*. The levels of QoxA, QoxB and QoxC were significantly reduced, while CoxB was significantly elevated in the spore inner membrane. On the other hand, the NADH dehydrogenase NUO (NuoH BC5297, Figure 5d) and cytochrome c reductase complex (BC1522 ISP, BC1523 Cyt b, BC1524 Cyt 1, Figure 5e) were only found in the spore inner membrane, while the cytochrome bd complex (CydA BC4792, Figure 5f) was significantly higher in the cell membrane fraction.

#### 2.3.3. Toxins and Virulence

In the group of *B. cereus sensu lato*, it is sometimes difficult to distinguish closely related species based on their genome sequences and these are then additionally classified according to their abilities to produce the parasporal toxin [26]. Vegetative cells of *B. cereus* can produce enterotoxins and membrane-damaging toxins. The enterotoxins causing diarrhea are hemolysin BL (HBL), non-hemolytic enterotoxin (NHE), cytotoxin K [27], which will accumulate during stationary phase at high cell density. The strain *B. cereus* ATCC 14579 used here does not produce the heat-stable emetic toxin cereulide [28]. In this study, several toxins (HblB BC3102, Hbl1 BC3103, NheA BC1809, NheB BC1810, EntFM BC1953, EntC BC0813) were identified (Figure 5g), except for EntFM, toxins were predominant in the cell membrane fraction. PlcR (Phospholipase C Regulator) is not a membrane protein [29] but is an important transcriptional activator protein controlling most known virulence factors like enterotoxins [29], which was identified in the cell membrane fraction. Two chemotaxis proteins (BC0576, BC3385) and one GGDEF-family regulator (BC3747) under PlcR regulation (Figure 5h) were identified predominantly in cell membrane preparations. All these three proteins are sensors of the 253 *B. cereus* PlcR regulon [29].

For *B. cereus*, virulence is regulated mainly via secretory complexes, alternative sigma factors or standalone transcriptional regulators which also respond to a changing environment [29]. One essential secretory system is the Sec system regulating virulence, which is also part of the quorum sensing pathway [30] and was identified in the spore inner membrane and cell membrane (Figure 5i). Another type of secretion system twin arginine translocation (Tat) system was not identified [31]. Of the Sec system, seven proteins (BC2740 SecY, BC4405 SecD/F, BC3843 ffh, BC4410 YajC, BC5488 YidC2, BC3070, BC3837) are part of the conserved post-translational translocation protein secretion system. SecY, SecD/F, YajC and YidC2 are the transmembrane channel. SPase I is an essential part of the Sec-dependent protein export pathway, especially for secreted virulence factors. It is responsible for cleaving of the signal peptide region from non-lipoprotein preproteins synthesized in the cytosol. In *B. subtilis,* five genes express SPase I(SipS, SipT, SipU, SipV, and SipW) [32]. However, *B. cereus* has six genes (BC0456, BC1136, BC2621, BC3060, BC3070, BC3837) of which BC3070, BC3837 were identified in both spore inner membrane and cell membrane, although expression in the cell membrane was higher.

### 2.4. Unique Proteins Detected in B. cereus

*B. subtilis* is the well-studied model organism of the genus *Bacillus*, whereas *B. anthracis* is closely related to *B. cereus* and can hardly be distinguished based only on genome analysis. Compared with previous reports [12,33], some proteins are worth noticing. An orphan kinase (BC5455, Figure 5j) which was reported to be a unique gene in *B. cereus*, absent from *B. subtilis* and *B. anthracis* [34], was only identified in vegetative cell membrane preparations. The specific role of BC5455 in *B. cereus* is unclear.

## 3. Discussion

The membrane forms the barrier between a cell’s interior and exterior. The mem-brane is a nexus for both perception of the environment as well as import of nutrients and export of signals and components that make up the extracellular parts of an organism. Detailed study of the composition of the membrane protein make-up shows us how various crucial processes at the cell membrane are organized during different stages of the bacterial life cycle of *B. cereus*. In the current study we used a membrane enrichment approach to isolate the cellular membrane of mid-log phase vegetative cells and the inner membrane of dormant spores of *B. cereus*. As is obvious, there is still a substantial number of proteins detected that would not be considered an integral part of the membrane nor membrane-associated (Figure 1d). This is a limitation of membrane enrichment, where only limited washing is feasible in order not to disrupt and remove the delicate membrane structures [12]. This contrasts with analysis of integral cell wall or spore coat proteins in eukaryotes [35] and prokaryotes [36], where the focus on covalently linked proteins allows for more unequivocal proof of localization, at the expense of proteins without covalent linkages.

This is a drawback of the current approach as it limits our ability to detect novel localization of proteins and necessitates further bioinformatic prediction of membrane localization to separate likely contaminants from bona-fide (integral) membrane proteins. This also makes us relatively insensitive to peripheral membrane proteins as these are less easily characterized by bioinformatic methods [37]. These proteins that can attach to lipids or integral membrane proteins in a reversible manner are also of great interest [38]. The enrichment of membrane fractions does give us an improved detection of membrane proteins compared to whole cell or spore proteomics and provides the first quantitative and most complete analysis of the inner membrane proteome of spores and cell membrane of vegetative cells of *B. cereus* to date.

The composition of the membrane proteome differs significantly between vegetative cells and dormant spores, and we identify twice as many cell membrane proteins from vegetative cell membranes (244 vs. 498, Figure 4), among which is a far larger number of various transporters, receptors and proteins related to cell motility and cell division compared to the spore inner membrane. Also, among the proteins which are shared between cells and spores, several transporters, cellular motility and environmental perception proteins were expressed at an elevated level in the vegetative cell membrane. This underscores the difference between stages of the *B. cereus* life cycle. A vegetative cell needs more types of transporters to provide its metabolism with sufficient nutrients and energy from various carbon sources and thus expresses transporters for different types. Simple carbohydrates like glucose and fructose seem to be preferred for providing energy during the early stages of germination, as the *B. cereus* spore waiting to germinate expresses a transporter for these carbohydrates.

Another membrane-bound pathway where differences in membrane protein com- position between vegetative cells and spores is apparent is energy metabolism. In the *B. cereus* spore inner membrane, two types of NADH dehydrogenases (type I and II) were identified, whereas only type II NADH dehydrogenase was present in vegetative cell membranes. Two alternative NADH dehydrogenases (NDH-I and NDH-II) exist in *Escherichia coli* for response to different environmental conditions. NDH-I, encoded by the *nuo* gene, is reported to be a more efficient NADH dehydrogenase and essential for the aerobic growth of *Shewanella oneidensis* in a minimal medium [39], and is absent in the genome of *B. subtilis* [40]. NDH-II is smaller, does not couple NADH oxidation with ion transport, and is encoded by *ndh* gene, whose expression level is reported to increase during exponential growth [40]. Also, cytochrome c reductase is only identified in the spore inner membrane, the function of which in the *B. cereus* spore is unclear. The dormant spore is assumed to be metabolically inactive, so perhaps the expression of both types of NADH dehydrogenases is important for energy metabolism during germination, whereas NDH-II is more important in an actively growing cell. The acetyl-coenzyme A synthetase (BC2489) may also play a role in dormant spores. Since it can convert acetate in low concentration to acetyl-CoA, this may help to keep the tricarboxylic acid (TCA) cycle in operation. The data underscore that *B. cereus* spores contain a set of proteins that can immediately after the first phases of germination be used for energy conversion processes under aerobic 335 conditions.

Consistent with previous studies that only vegetative cells have the ability of producing toxins, the proteins that are involved in biosynthesis and regulation of toxins were found in the vegetative cell membrane. Of note is that the enterotoxin EntFM (BC1953) was found both in spore and cell membrane preparations, different from other toxins which were predominant in the cell membrane fraction. Although called a toxin, it is reported to possibly be a cell wall peptidase according to sequence homology and secondary structure analysis, involved in bacterial motility and shape, adhesion to epithelial cells, and biofilm formation in *B. cereus* [41]. While EntFM itself is not a toxin, its function in steering bacterial cell wall dynamics contributes to bacterial pathogenicity.

Many details of the process of germination in *B. cereus* are based on genome analysis [42] and on the more well-studied model organism for Gram-positive bacteria *B. subtilis*. Spore-specific proteins that reside in the membrane are the germinant receptors and other spore proteins of which 20 were identified (Table 1). The subunit SpoVAE (BC4065) was identified but predicted not to be a membrane protein, even though in *B. subtilis* SpoVAEa and SpoVAEb were reported to be localized in the inner membrane. Here we quantify their abundance in the membrane and show reasonable correlations between the expression ratios of various components and earlier studies in *B. subtilis* [19] and *B. cereus* [18]. Our study here shows the first global quantitation of several the subunits of the “germinosome” in the spore membrane of *B. cereus* and so can give an indication of how the stoichiometry of such a complex could be organized in the spore membrane.

Overall, the current study gives an insight into the quantitative composition of the membrane proteome of cells and spores of *B. cereus* in detail. The current approach is a starting point for future studies into how membrane composition changes during different growth phases, and morphological forms in response to the environment of Gram-positive spore-forming bacteria.

## 4. Materials and Methods

### 4.1. Strain and Culture Conditions

The strain *B. cereus* ATCC 14579 was used in this study. A single colony was inoculated in liquid tryptic soy broth (TSB) medium, grown aerobically at 30 °C and 200 rpm. Then, cells cultured overnight were transferred to a chemically defined growth and sporulation (CDGS) medium [43] and incubated for 96 h before spores were harvested. Spores were harvested by centrifugation with 0.1% Tween-80 and washed with cold Milli-Q water for at least four times. Vegetative cells were collected at early exponential phase.

### 4.2. Isolation of Spore Inner Membrane and Vegetative Cell Membrane

The isolation method of the spore inner membrane was derived from the isolation of the inner membrane from *B. subtilis* spores [12]. In brief, approximately an OD_600_ of 80 (100–150 mg dry weight) of purified spores was treated by the following steps. First, spores were decoated by a treatment with 0.1 M NaCl/0.1 M NaOH/1% sodium dodecyl sulfate (SDS)/0.1 M dithiothreitol (DTT) at 70 °C for 1 h. After intensive washing with water, decoated spores were incubated with TEP buffer (50 mM Tris-HCl (pH 7.4), 5 mM EDTA, 1 mM phenylmethylsulfonyl fluoride) containing 2 mg/mL lysozyme and 40 µg/mL MgCl_2_ at 37 °C for 30 min. The spores were intensively washed with TEP and then disrupted by bead-beating with 0.1 mm Zirconia-Silica beads (BioSpec Products, Bartlesville, OK, USA) using a Precellys 24 homogenizer (Bertin Technologies, Aix en Provence, France) (six rounds of 20 s at 6000 rpm with ice cooling between each round). The beads were rinsed with TEP buffer several times, and the suspension was collected and centrifuged at 15,000 rpm for 5 min at 4 °C to remove unbroken cells and spores. NaCl was added at a final concentration of 1 M to the supernatant, which contained the membrane and soluble fractions. The supernatant was ultracentrifuged at 100,000× *g* for 1 h at 4 °C. The pellet was washed with carbonate buffer (100 mM Na_2_CO_3_, 100 mM NaCl, 10 mM EDTA, pH 11) at 4 °C on a shaker for 1 h and ultracentrifuged as mentioned earlier, producing the final pellet of IM. The membrane fractions of an OD_600_ of 20 of vegetative cells were pelleted using the same method as above. The samples of spore inner membrane and vegetative cell membrane were collected from three biological replicates.

### 4.3. Sample Preparation for Proteomics Analysis

Membrane pellets were resuspended in 200 µL of lysis buffer (6 M urea, 5 mM DTT, 50 mM NH_4_HCO_3_, pH = 8.0) and incubated at 55 °C for one hour to reduce disulfide bridges. Then samples were treated with 15 mM iodoacetamide (IAA) to alkylate free cysteines in the dark for 45 min. Alkylation was quenched by addition of 20 mM thiourea. Next, samples were diluted six times with 50 mM NH_4_HCO_3_, and digested by trypsin (1:100 w/w protease/protein ratio) at 37 °C for 18 h. The mixture of peptides was freeze-dried and dissolved in 0.1% trifluoroacetic acid (TFA) followed by clean-up using C18 reversed-phase TT2 Top-Tips (Glygen). The final peptide fractions were suspended with 0.1% FA for MS analysis. The samples of whole cell and spores were prepared as described above.

### 4.4. LC-MS/MS Analysis

Mass spectrometric analysis of 200 ng peptides was carried out on a timsTOF pro (Bruker, Bremen, Germany) equipped with an Ultimate 3000 nanoRSLC UHPLC system (Thermo Scientific, Germeringen, Germany). Samples were injected onto a C18 column (75 µm, 250 mm, 1.6 µm particle size, Aurora, Ionopticks, Fitzroy, Australia) kept at 50 °C. Peptides were loaded at 400 nl/min for 2 min in 3% solvent B and separated by a multi-step gradient: 6% solvent B for 55 min, 21% solvent B for 21 min, 31% solvent B for 12 min, 42.5% solvent B for 3 min and 99% solvent B for 7 min (Solvent A: 0.1% formic acid in water, Solvent B: 0.1% formic acid in acetonitrile). MS analysis of eluting peptides was performed by a time-of-flight mass spectrometer with collision energy from 20–59 eVa. The precursor scan ranged from 100 to 1700 m/z and a tims range of 0.6–1.6 V.s/cm2 in PASEF mode. A total of 10 PASEF MS/MS scans were collected with a total cycle time of 1.16 s.

### 4.5. Data Analysis

Raw MS/MS data were processed using Maxquant software (version 1.6.14.0) [44], searching a proteome database of *B. cereus* (Uniprot, downloaded 1-9-2019), to estimate false spectrum assignment rate a reverse version of the same database was also searched. The settings were as follows: Enzyme Trypsin/P allowing for a maximum of 2 missed cleavages, variable modifications: Oxidation (M), fixed modifications: Carbamidomethyl (C). Settings were default for timsDDA, match between runs was enabled with a matching time window of 0.2 minutes and a matching ion mobility window of 0.05 indices. For label-free quantification, both iBAQ and LFQ were enabled [44].

Proteins that were identified for at least two replicates were kept for predictions of membrane localization by the LocateP [9], PSORTb [10] and TMHMM [11] algorithms. The membrane proteins contain multitransmembrane, multitransmembrane (lipid-modified N termini), lipid anchored, LPxTG cell wall anchored, N-terminally anchored (no cleavage site), N-terminally anchored (with cleavage site), C-terminally anchored (with cleavage site), intracellular/TMH start after 60) predicted by LocateP. Proteins predicted to be in “CytoplasmicMembrane” by PSORTb or to be harboring at least one transmembrane domain by TMHMM were also classified to be membrane proteins.

Predicted membrane proteins from the spore inner membrane and cell membrane fractions were analyzed in Perseus (version 1.6.15.0) [45] using iBAQ intensity. The functions of proteins were categorized according to Gene Ontology and KEGG pathway. The iBAQ intensity of predicted membrane proteins that were shared between cell membrane and spore inner membrane was used for a volcano plot using a T-test for determining significant changes of protein levels. The p-values were adjusted for multiple testing using a permutation-based FDR to obtain an FDR of 0.01 (as implemented in Perseus).

Homologues of uncharacterized proteins in other bacteria were detected using the Basic Local Alignment Search Tool (BLAST: BLASTP 2.9.0+) at uniprot.org, (https://www.uniprot.org/blast/, accessed on 8 November 2021). The settings used were as follows, Matrix: blosum62, Threshold (E-value): 10, Filtering for low complexity regions turned on, Gapped: True, max number of hits reported: 100. We used the uniprotkb bacteria (Protein) generated for BLAST on 16 June 2021 with 151,792,219 sequences consisting of 48,030,169,589 letters to search for homologues. A selection of the 3 highest scoring homologues in other species was made, with at least 50% identity, prioritizing proteins that were not listed as uncharacterized. Complete blast results can be found in Appendix A.

## Figures and Tables

**Figure 1 ijms-22-12475-f001:**
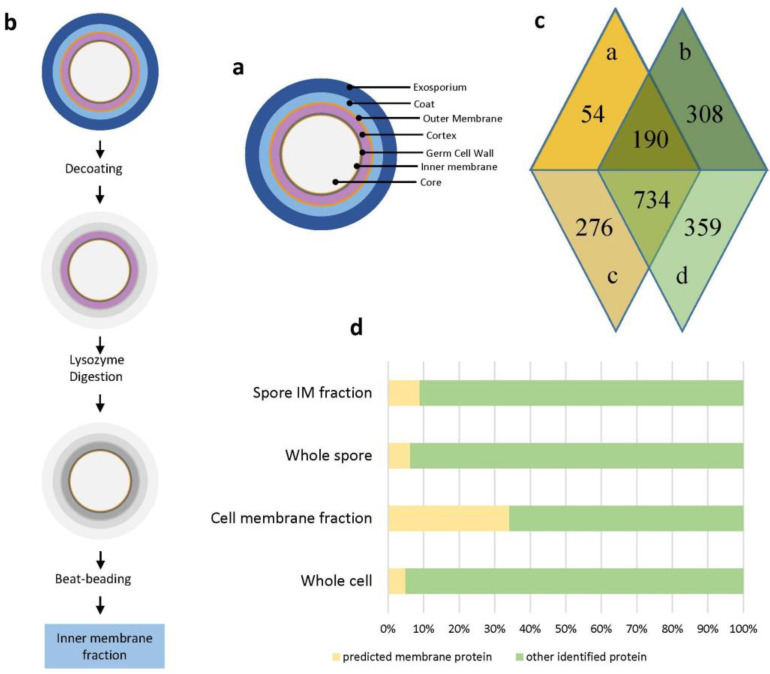
Spore structure, membrane enrichment and identified proteins from *B. cereus* spore inner membrane and vegetative cell membrane. (**a**) Spore structure. (**b**) Workflow for inner membrane enrichment. (**c**) Predicted localization of proteins identified from membrane factions of *B. cereus* spores and cells. (Triangle a) the number of predicted spore inner membrane proteins; (Triangle b) the number of predicted cell membrane proteins; (Triangle c) the number of other spore proteins identified in spore inner membrane fraction; (Triangle d) the number of other cell protein identified in cell membrane fraction. (**d**) Percentage of predicted membrane protein levels in each group.

**Figure 2 ijms-22-12475-f002:**
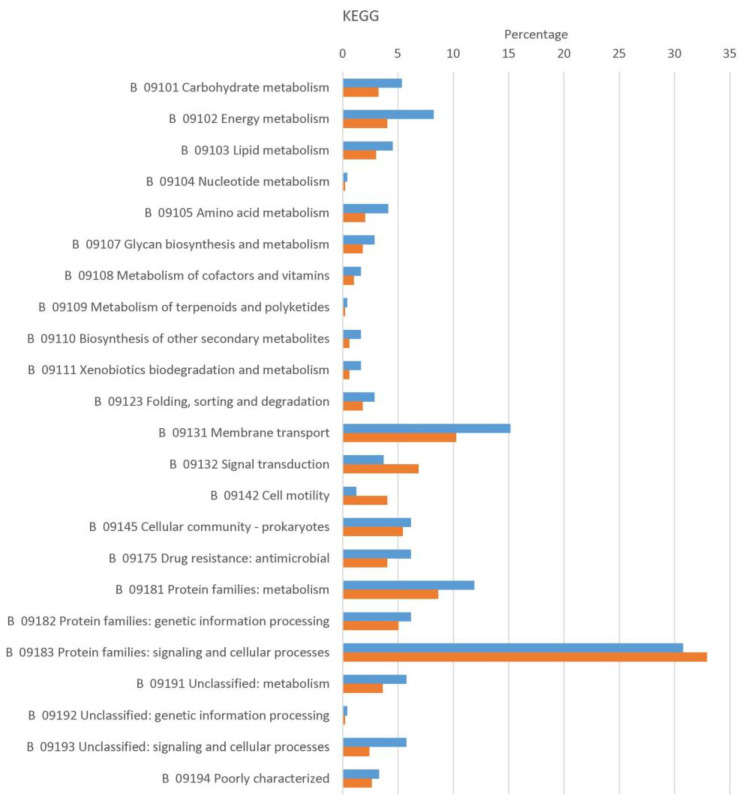
Functional categories of membrane proteins identified in spore inner membrane and cell membrane of *B. cereus*. Involvement in various KEGG pathways. The percentage of the total number of proteins (spores: 244, cells: 498) assigned to KEGG terms are shown for cells (orange) and spores (blue).

**Figure 3 ijms-22-12475-f003:**
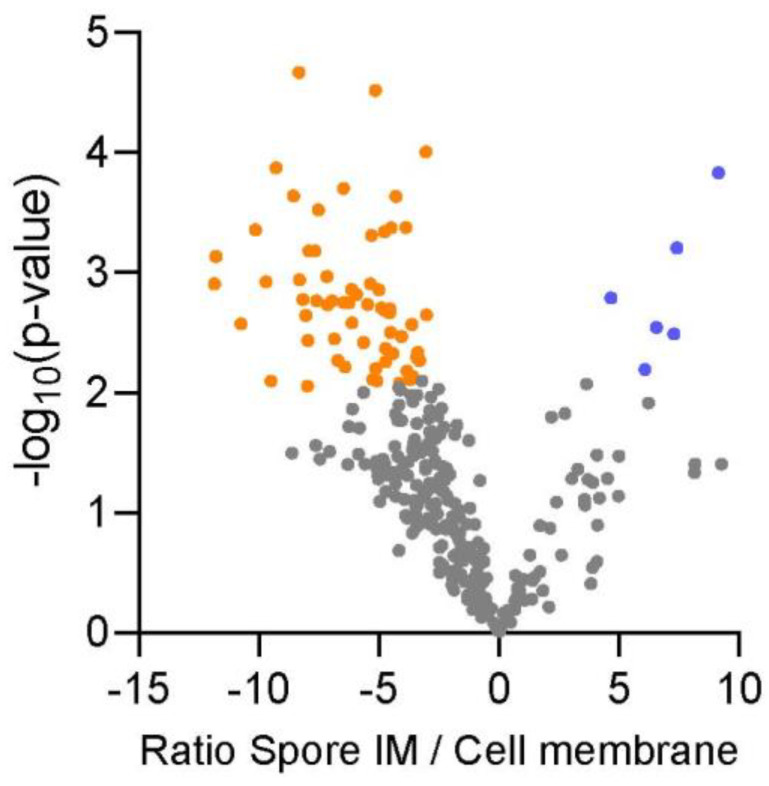
Comparison of spore inner membrane proteins and vegetative cell membrane proteins. Volcano plot of the ratio of iBAQ-values of the spore inner membrane proteins and vegetative cell membrane proteins that were found in both sample types. Significantly more abundant proteins (FDR *≤* 0.01) in the cell membrane (orange) and spore inner membrane (blue) are shown by colored circles.

**Figure 4 ijms-22-12475-f004:**
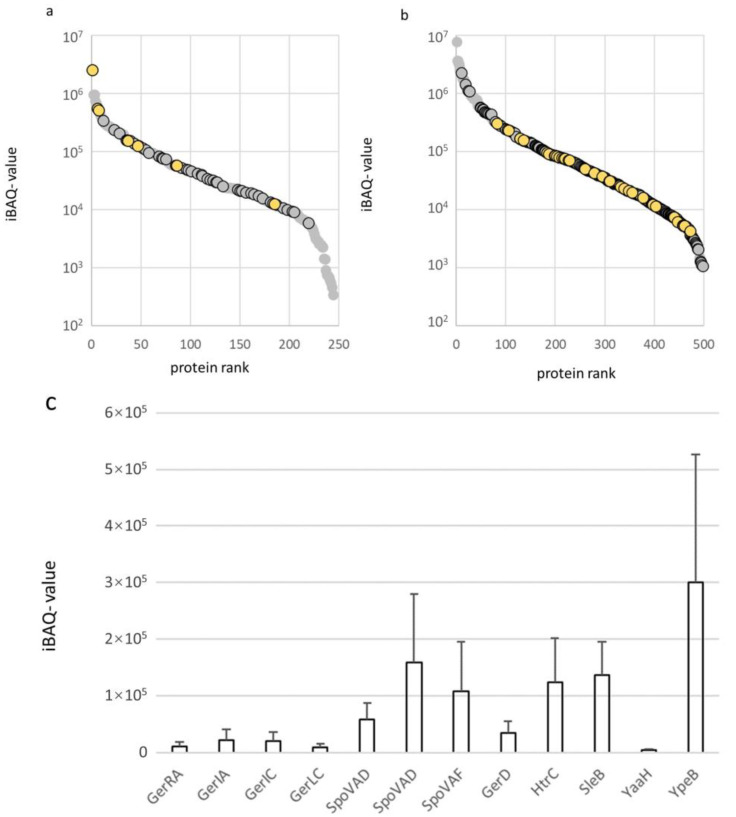
Abundance of membrane annotated proteins in the cell and spore membrane fractions. Membrane annotated proteins are shown ranked according to iBAQ values for spore inner membrane (**a**) and cell membrane preparations (**b**). Circles with a solid border mark protein exclusively identified in spores and cells, yellow circles mark uncharacterized proteins. (**c**) Proteins related to germination of bacillus spores, mean iBAQ values of 3 replicates are shown, error bars denote the standard error of the mean.

**Figure 5 ijms-22-12475-f005:**
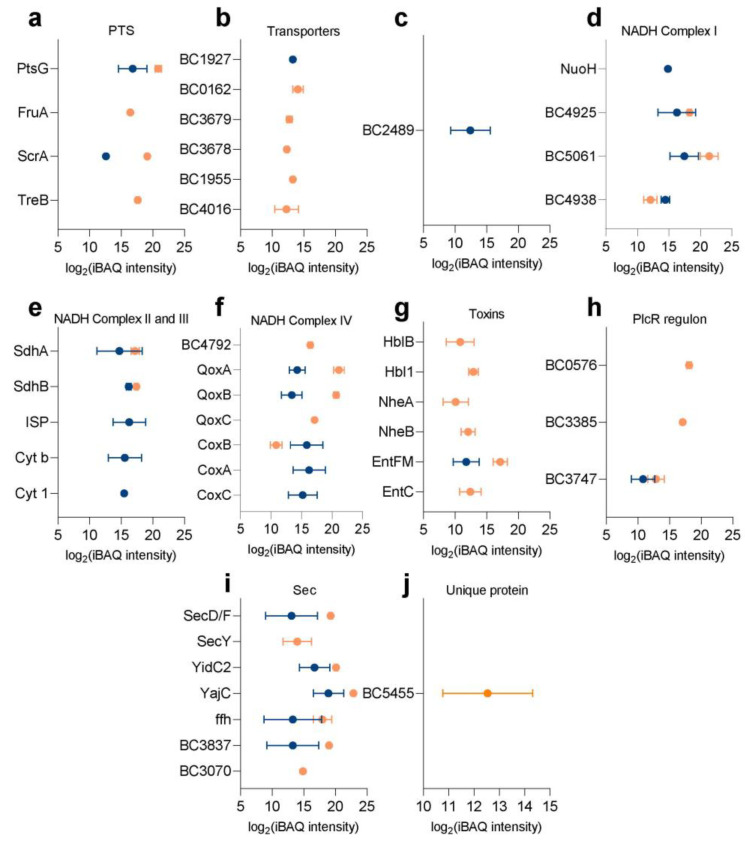
Abundances of proteins with vital functions in the cell or spore membrane. The iBAQ values of proteins from spore inner membrane (blue) and from the vegetative cell membrane (orange) associated with PTS (**a**), transporters (**b**), acetyl-coenzyme A synthetase (**c**), NADH complex I (**d**), NADH complex II and III (**e**), NADH complex IV (**f**), toxins (**g**), PlcR regulon (**h**), Sec system (**i**) and a unique protein (**j**) are shown, see text for details.

**Table 1 ijms-22-12475-t001:** Germination associated proteins identified in this study.

Protein	Gene	Accession No.	N †	Description
GerKA	BC_0635	Q81HZ1	1	spore germination protein KA
GerKC	BC_0633	Q81HZ3	1	spore germination protein KC
GerSA	BC_3574	Q81AJ2	1	spore germination protein SA
GerRA	BC_0784	Q81HM0	2	GerA family spore germination protein
GerRC	BC_0783	Q81HM1	2	GerC family spore germination protein
GerQC	BC_3097	Q81BQ4	2	spore germination protein QC
GerIA	BC_4731	Q816T6	2	spore germination protein IA
GerLC	BC_0706	Q81HS6	2	spore germination protein LC
SpoVAA	BC_4070	Q819B6	2	stage V sporulation protein AA
SpoVAC	BC_5147	Q815K0	2	stage V sporulation protein AC
SpoVAD	BC_4067	Q819B9	2	stage V sporulation protein AD
SpoVAD	BC_5148	Q815J9	3	stage V sporulation protein AD
SpoVAE	BC_4065	Q819C1	1	stage V sporulation protein AE
SpoVAF	BC_4064	Q819C2	3	stage V sporulation protein AF
GerD	BC_0169	Q81J09	3	spore germination protein GerD
HtrC	BC_5458	Q814H6	3	serine protease YyxA
SleB	BC_2753	P0A3V0	3	Spore cortex-lytic enzyme
YaaH	BC_3607	Q81AG3	3	spore peptidoglycan hydrolase
YpeB	BC_2752	Q813I5	3	hypothetical protein (sporulation protein)
† Protein identified in N number of replicates out of 3.				

## Data Availability

Mass spectrometry data have been deposited and can be found at ProteomeXchange (PXD029025), and the Massive Repository for Mass Spectrometry data (MSV000088208).

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
