# Peer review of "The Membrane Proteome of Spores and Vegetative Cells of the Food-Borne Pathogen Bacillus cereus"

_ijms, 2021, doi:10.3390/ijms222212475_

Round 1

Reviewer 1 Report

The authors introduce  a methodology to quantitatively assess the membrane proteome of cells and spores of B. cereus- a human pathogen known to spread food-borne diseases. They introduce an approach to identify and quantify the subunits of germinosome in the spore membrane of this gram positive bacteria. They present the relative abundance of different proteins in terms of iBAQ values using bioinformatics approach combined with cell membrane enrichment followed by label free mass spectrometry.

Thank you for sharing your interesting work to understand the fundamental difference in the spore and cell membrane protein composition driving sporulation / germination of this human pathogen. This understanding can contribute to mitigation of the spread of this bacteria which causes food-borne diseases. Please find some comments to be addressed below-

  • The authors mention that the membrane fraction enrichment limits the specificity of this method such that not all detected fractions are membrane associated proteins- have you considered using detergent based method to selectively extract membrane proteins(Feroz et al, Analyst 2017)
  • Some general comments: please expand the acronyms when using the in text first time such as GO, KEGG and TCA.
  • The iBAQ values clearly show greater abundance of membrane proteins in the vegetative cell membrane than spore form. The authors present BC2489 as unique to spores to maintain the TCA cycle- can they elaborate on the possible causes of difference in the levels for this protein in the spore versus cell membrane? 
  • Similar comment for the orphan kinase in the same figure which has greater abundance on the cell membrane...

Author Response

Dear Editor,

The authors would like to thank the reviewers for the insightful and constructive comments. The changes to the manuscript that were made, markedly improved the quality of the manuscript. In addition to these changes, we have responded to the comments of the reviewers below.

Kind Regards, on behalf of all authors

Gertjan Kramer

Reviewer 1

Comment 1: The authors mention that the membrane fraction enrichment limits the specificity of this method such that not all detected fractions are membrane associated proteins- have you considered using detergent based method to selectively extract membrane proteins (Feroz et al, Analyst 2017)

Response: We try to analyze the complete spore inner membrane proteome including integral, peripheral and lipid-anchored proteins. There is no detergent that is suitable for purification of all membrane proteins of different properties. Using a certain detergent results in a loss of information. What’s more, a high concentration of detergent (several times critical micelle concentration (CMC)) used in purification of integral membrane proteins requires a remove of detergent before analysis by mass spectrometry, which needs extra steps and efforts. Membrane proteins have the tendency to aggregate, which may reduce the efficiency of digestion and detection by mass spectrometry. However, without detergents, the efficiency of membrane protein enrichment reduced in our experiment. But the reviewer provides a very nice method to concentrate low abundant and extremely hydrophobic membrane proteins, which may be considered to use in further study of specific spore inner membrane proteins, like the germinant receptors.

Comment 2: please expand the acronyms when using the in text first time such as GO, KEGG and TCA.

Response: As suggested by the reviewer, we have revised the acronyms mentioned in the text (line 92, 93, 332).

Comment 3: The iBAQ values clearly show greater abundance of membrane proteins in the vegetative cell membrane than spore form. The authors present BC2489 as unique to spores to maintain the TCA cycle- can they elaborate on the possible causes of difference in the levels for this protein in the spore versus cell membrane?

Similar comment for the orphan kinase in the same figure which has greater abundance on the cell membrane...

Response: One publication (Brillard, Julien, et al. "Identification of Bacillus cereus genes specifically expressed during growth at low temperatures." Applied and Environmental Microbiology 76.8 (2010): 2562-2573.) reports that BC2489 is specifically expressed during growth at low temperatures. A hypothesis is that this protein is related to stress response and is expressed during sporulation.

There is only one publication (Anderson, Iain, et al. "Comparative genome analysis of Bacillus cereus group genomes with Bacillus subtilis." FEMS Microbiology Letters 250.2 (2005): 175-184.) that mentions the orphan kinase. The function and regulation of this protein are unclear. The reason why this kinase is predominant in cell membrane is unclear to us.

Reviewer 2 Report

In Line 240 and 244 was reported the same sentence: The strain B. cereus ATCC 14579 does not produce the heat-stable emetic toxin cereulide.

 I'd like to suggest a more recent work to the authors. For example: The strain B. cereus ATCC 14579 does not produce the heat-stable emetic toxin cereulide. Also in wild strains few evidence were reported about toxin cereulide detection, as reported by  (Montone AMI, Capuano F, Mancusi A, et al. Exposure to Bacillus cereus in Water Buffalo Mozzarella Cheese. Foods. 2020;9(12):1899. Published 2020 Dec 19. doi:10.3390/foods9121899) the cereulide gene was detected in only one B. cereus isolate, jointly with the complete nhe operon and entFM gene

Author Response

Dear Editor,

The authors would like to thank the reviewers for the insightful and constructive comments. The changes to the manuscript that were made, markedly improved the quality of the manuscript. In addition to these changes, we have responded to the comments of the reviewers below.

Kind Regards, on behalf of all authors

Gertjan Kramer

Reviewer 2

Comment 1: In Line 240 and 244 was reported the same sentence: The strain B. cereus ATCC 14579 does not produce the heat-stable emetic toxin cereulide.

Response: The repeated sentence has been removed.

Comment 2. I'd like to suggest a more recent work to the authors. For example: The strain B. cereus ATCC 14579 does not produce the heat-stable emetic toxin cereulide. Also in wild strains few evidence were reported about toxin cereulide detection, as reported by (Montone AMI, Capuano F, Mancusi A, et al. Exposure to Bacillus cereus in Water Buffalo Mozzarella Cheese. Foods. 2020;9(12):1899. Published 2020 Dec 19. doi:10.3390/foods9121899) the cereulide gene was detected in only one B. cereus isolate, jointly with the complete nhe operon and entFM gene.

Response:  Thank you for suggesting this paper to us, we have read it with great interest.